# The Structural Role of Smart Contracts and Exchanges in the Centralisation of Ethereum-Based Cryptoassets

**DOI:** 10.3390/e24081048

**Published:** 2022-07-30

**Authors:** Francesco Maria De Collibus, Matija Piškorec, Alberto Partida, Claudio J. Tessone

**Affiliations:** 1Blockchain and Distributed Ledger Technologies Group, Universität Zürich, Andreasstrasse 15, CH-8050 Zürich, Switzerland; 2Rudjer Boskovic Institute, Bijenicka cesta 54, 10000 Zagreb, Croatia; matija.piskorec@irb.hr; 3International Doctoral School, Móstoles Campus, Rey Juan Carlos University, 28933 Madrid, Spain; apartidar@gmail.com; 4UZH Blockchain Center, Universität Zürich, Andreasstrasse 15, CH-8050 Zürich, Switzerland; claudio.tessone@uzh.ch

**Keywords:** blockchain, Ethereum, decentralised finance, network science, preferential attachment, network dismantling

## Abstract

In this paper, we use the methods of networks science to analyse the transaction networks of tokens running on the Ethereum blockchain. We start with a deep dive on four of them: Ampleforth (AMP), Basic Attention Token (BAT), Dai (DAI) and Uniswap (UNI). We study two types of blockchain addresses, smart contracts (SC), which run code, and externally owned accounts (EOA), run by human users, or off-chain code, with the corresponding private keys. We use preferential attachment and network dismantling strategies to evaluate their importance for the network structure. Subsequently, we expand our view to all ERC-20 tokens issued on the Ethereum network. We first study multilayered networks composed of Ether (ETH) and individual tokens using a dismantling approach to assess how the deconstruction starting from one network affects the other. Finally, we analyse the Ether network and Ethereum-based token networks to find similarities between sets of high-degree nodes. For this purpose, we use both the traditional Jaccard Index and a new metric that we introduce, the Ordered Jaccard Index (OJI), which considers the order of the elements in the two sets that are compared. Our findings suggest that smart contracts and exchange-related addresses play a structural role in transaction networks both in DeFi and Ethereum. The presence in the network of nodes associated to addresses of smart contracts and exchanges is positively correlated with the success of the token network measured in terms of network size and market capitalisation. These nodes play a fundamental role in the centralisation of the supposedly decentralised finance (DeFi) ecosystem: without them, their networks would quickly collapse.

## 1. Introduction

Blockchain is a decentralised technology that stores a list of transactions in a distributed way while keeping their integrity [1]. The pseudonym Satoshi Nakamoto in 2008 introduced this new approach to solve the double spending problem in the paper: “Bitcoin, a peer-to-peer electronic cash system” [2]. In 2013, Vitalik Buterin proposed [3], and soon developed, together with Gavin Wood et al., Ethereum, a public blockchain that included a Turing-complete computing platform using *smart contracts*: programs that are stored and executed on the blockchain via *decentralised applications* (Dapps). Smart contracts gave birth to a myriad of cryptoassets including *tokens*, both *fungible* (FTs) and *non-fungible* (NFTs). Many of those FTs are instrumental for decentralised finance (DeFi) applications. Ethereum uses an *account-based model* with balances associated to each blockchain address, while Bitcoin uses the *unspent transaction outputs* (UTXO) model [3] and provides a limited extent of “programmability” with the Bitcoin Script, a stack-based Turing-incomplete language.

The Ethereum Request for Comments 20 (ERC-20) [4] is the Ethereum standard for fungible (identical and interchangeable) non-native tokens. This specification defines an interface, a set of methods such as *transfer()*, *balanceOf()* and *approve()* to be implemented within the smart contract that defines the token. In addition to the native cryptocurrency Ether, there were more than 500k tokens on Ethereum by June 2022 [5]. Cardano, Solana, Dfinity and Moonbeam in the Polkadot [6] network are later examples of public blockchains that support smart contracts. In this study, we analyse Ethereum-based tokens through the network science lens. We focus on the transaction networks of a set of tokens and Ether We compare them with each other to understand whether different cryptoassets present different properties or growth patterns.

Transactions occur between addresses in the Ethereum network. Due to the use of the account model, each address has an associated balance. In a transaction, the sender address sends data, which can be a specific amount of value or raw data such as a function call, to the destination address. Each address in Ethereum belongs to an *externally owned account* (EOA) or to a *smart contract* (SC). An EOA has an address derived from a public key, uniquely associated to a private key, which can be owned by a person or an off-chain program. SCs also have an address, and they expose functions that can be invoked by EOAs or other contracts. As SCs do not have their own key pair composed of a public and a private key, they cannot initiate a chain of transactions themselves [3,7]. We create networks whose nodes are the addresses, and the edges are the transactions between those addresses. We use the term “node” here as a synonym for address; we will never refer to the Ethereum clients running on the individual computers in this article.

The Ether transaction network in Ethereum has been studied extensively: Bai et al. [8] in 2022 used a temporal network perspective. Kondor et al. [9] in 2021 performed a comparative study between the Ethereum transaction network and other cryptocurrencies, with a deep dive on the Matthew Effect of concentration (“the rich get richer”). More recently, Campajola et al. [10] in 2022 analysed the centralisation of different cryptocurrencies. These studies, however, do not focus on non-native tokens. Somin et al. [11] in 2020 studied Ether and ERC-20 networks, finding scale-free properties, but the authors did not distinguish between individual networks of different tokens. Focused on tokens, Victor and Lüders [12] in 2019 analysed instead the transaction networks of many ERC-20 tokens. They found that not all transaction networks are scale-free. However, they used transaction data from 2018, before the big wave of adoption and the crypto market exuberance of early 2021. Chen et al. [13] in 2020 studied the relation between token creators and token holders and how they interact with a popular exchange. They proposed an algorithm to identify multiple addresses controlled by one entity.

ERC-20 tokens on top of the Ethereum blockchain constitute a unique object of study: their transaction networks interact among themselves and with the Ether transaction network. We started to study a limited set of ERC-20 tokens in 2021 [14]. In this article, we study the most connected addresses in the Ether transaction network and in specific token transaction networks. We first provide a deep dive on four DeFi-relevant (Decentralised Finance) ERC-20 tokens (Ampleforth, Basic Attention Token, DAI and Uniswap), and, second, a more general analysis across tokens. We analyse the structural properties of the transaction networks of Ether and ERC-20 tokens to understand whether all nodes (addresses) are equally relevant for the existence of the network. We complement previous studies with our analysis of the complete Ethereum blockchain until block 12,500,000 (24 May 2021). We conclude that smart contracts and exchanges play a structural role within the transaction networks in DeFi and Ether.

Cryptocurrency exchanges allow users to trade cryptocurrencies for other assets. These assets can be fiat money or other cryptocurrencies, both native or non-native tokens. Trades between cryptoassets are called swaps. Exchanges connect public blockchains with traditional means of payments such as credit cards and bank wires when they trade a cryptocurrency for fiat money. Custodial exchanges hold the private keys of their customers’ cryptoassets. In non-custodial exchanges, customers hold the private keys to their assets. From the blockchain perspective, the address of the custodial exchange is the owner of all their customers’ cryptoassets: internal trading is in this case mostly executed off-chain on the exchange’s balance sheets to save the cost of the transaction fee. Exchanges are also centralised in one or a handful of addresses or decentralised through one or multiple smart contracts. The presence of an exchange in a token transaction network signals interest from the market to trade that token.

Decentralised finance (DeFi) moves typical financial activities, such as lending and borrowing, from a traditionally centralised and intermediary-based scenario to a distributed one enabled by a blockchain. In DeFi, smart contracts run the logic required by these financial processes. At the time of writing, the Ethereum blockchain is the main DeFi platform [15].

The rest of this paper is organised as follows: we describe the data that we use for our analysis and the three elements of our methodology to study the role of specific participants in Ether and ERC-20 token transaction networks. We provide the results of our analysis and conclude with insights on how these networks display signs of centralisation.

## 2. Materials and Methods

### 2.1. Four Tokens Used as DeFi Collateral

This study is composed of a deep dive using four DeFi-relevant tokens and of a multi-faceted analysis of ERC-20 tokens in general that uses, first, multilayer networks and, second, broad similarity measures. Regarding the deep dive, the four tokens that we study belong to four distinct categories in DeFi: a *utility* token (BAT), an *algorithmic stablecoin* (AMP), a *multi-currency pegged algorithmic stablecoin* (DAI) and a *governance* token (UNI). Table 1 presents the data sets used for this analysis.

The following paragraphs describe the four tokens analysed in our deep dive:

**Ampleforth (AMP)**: An algorithmic stablecoin pegged to the US Dollar (USD). It achieves its stability by adapting its supply to price changes without a centralised collateral. The protocol receives exchange rate information from trusted oracles on USD prices and changes the number of tokens that each user holds automatically [16]. AMP was launched in June 2019 and had a market capitalisation close to USD 90M in May 2022 [17].

**Basic Attention Token (BAT)**: A utility token designed to improve efficiency in digital advertising via its integration with the Brave web browser. Users are awarded BAT tokens for paying attention to online advertisements. BAT’s value proposition allows users to maintain control over quantity and type of the advertisements they consume, while advertisers benefit from better user targeting and reduced fraud rates [18]. BAT had an initial coin offering (ICO) in May 2017, and as of May 2022 it had a market capitalisation close to USD 600M which placed it among the top 100 cryptocurrencies [17].

**Dai (DAI)**: A multi-currency pegged algorithmic stablecoin token [19] launched in 2017 which uses, as AMP, *smart contracts* on the Ethereum network to keep its value as close as possible to the US dollar. Users can deposit ETH as a collateral and obtain a loan in DAI. The stability of DAI is achieved by controlling the type of accepted collateral, the collaterisation ratio and interest rates. In November 2019, DAI transitioned from a single-collateral model (ETH) to a multi-collateral model with many more tokens and stablecoin, some of them, such as BAT and UNI, are analysed in this paper. As of May 2022 DAI had a market capitalisation close to USD 6B, which placed it within the top 20 cryptocurrencies [17].

**Uniswap (UNI)**: A decentralised finance protocol [20] to exchange ERC-20 tokens on the Ethereum network. Unlike traditional exchanges, it does not have a central, limited order book but rather a liquidity pool: pairs of tokens provided by users (liquidity providers) which other users can then buy and sell. This UNI governance token was launched on September 2020 [20]. This token allows its holders to take part in important decisions regarding Uniswap, and to own a share of the common UNI treasury. It is currently ranked among the top 30 cryptocurrencies with a market capitalisation close to USD 4B in May 2022) [17].

### 2.2. Transaction Networks in Blockchain

Public blockchains allow the scrutiny of all their transactions recorded in their ledger. Unlike in the traditional financial system, researchers have access to all transactions. Kondor et al. [21] in 2014, Liang et al. [22] in 2018, Bovet et al. [23] in 2019, Vallarano et al. [24] in 2020 and Somin et al. [11] also in 2020 applied networks science [25] to investigate the properties of the transaction networks underpinning cryptocurrencies.

The nodes in these transaction networks correspond to addresses. The edges correspond to the value transfers between them, each of which is stored as a distinct *transaction* on the Ethereum blockchain. Our analysis is inspired by our previous work on the role of smart contracts in transaction networks of DeFi-collateral Ethereum-based tokens [26]. We use network analysis considering the two types of nodes present in the transaction networks of the ERC-20 tokens: externally owned accounts (EOA) and smart contracts (SC). We study degree [27], density and largest strongly connected components. Along with these properties, we investigate preferential attachment and network dismantling.

Preferential attachment helps us understand network growth as new nodes and edges are added to the transaction network. It also explains wealth accumulation phenomena when the rich get richer: this is generically known as the “Matthew effect”, Yule’s effect or Gibrat’s Law. Linear preferential attachment is a model that assumes that new nodes preferentially attach to pre-existing nodes with higher degree, producing a scale-free network that displays a power-law degree distribution. Network dismantling, the opposite to network percolation, provides insights on how the network endures the elimination of a specific set of nodes [28,29,30].

### 2.3. Deep Dive: Network Construction

For each of the four Ethereum tokens, we construct an aggregated transaction network $G^{S}\left(t\right)$ as a directed graph containing the complete history of transactions for that particular token during the time span outlined in Table 1. The nodes in the network correspond to addresses participating in transactions, while each edge corresponds to all transactions between two addresses. In this network, all transactions occurring between two nodes are collapsed into one directed edge.
$$\begin{array}{c} G^{S}\left(t\right)=\left(V^{S}\left(t\right),E^{S}\left(t\right)\right)\mathrm{for}\;\mathrm{token}\;S\in \left\{\mathrm{AMP},\mathrm{BAT},\mathrm{DAI},\mathrm{UNI}\right\} \end{array}$$

The set of nodes $V^{S}\left(t\right)$ corresponds to the addresses that have been included in at least one transaction of token *S* since time *t*. The set of edges $E^{S}\left(t\right)$ consists of unweighted, directed edges between all pairs of addresses. Each transaction or set of transactions between two nodes is represented as an edge $\left(j_{1},j_{2}\right)$ where node $j_{1}$ is the sender and node $j_{2}$ is the recipient.

### 2.4. Preferential Attachment

In the preferential attachment model, the probability π that a new node forms a link with an existing node *j* is directly proportional to the in-degree $k_{in,j}$ of the node *j*. This can be expressed in the following way:(1)$$\pi \left(k_{in,j}\right)=\frac{{\left(k_{in,j}\right)}^{\alpha^{in}}}{\sum_{j^{\prime }}{\left(k_{in,j^{\prime }}\right)}^{\alpha^{in}}},$$

The preferential attachment coefficient $\alpha^{in}>0$ determines the linearity of the attachment. A value of $\alpha^{in}=1$ implies that the preferential attachment is linear, while a value of $\alpha^{in}<1$ or $\alpha^{in}>1$ implies, respectively, a sub-linear or a super-linear attachment. A linear probability to form a link leads to a scale-free network. On the other hand, super-linear probability results in a network where very few nodes tend to connect to all other nodes of the network, effectively acting as super-hubs in the network. This model was used extensively for in-degree in an article on preferential attachment in key Ethereum-based cryptoassets published in 2021 by the authors of this study [14]. In this case, we extend this preferential attachment model to the out-degree $k_{out,j}$ of an existing node *j* to model the dynamic process of out-degree consolidation.
(2)$$\pi \left(k_{out,j}\right)=\frac{{\left(k_{out,j}\right)}^{\alpha^{out}}}{\sum_{j^{\prime }}{\left(k_{out,j^{\prime }}\right)}^{\alpha^{out}}}.$$

When a new directed edge is added to the network from an existing node, we assume that this source node *j* is selected with a probability $\pi \left(k_{out}^{*}\right)$, which is solely a function of its out-degree $k_{out}^{*}$. As the probability $\pi \left(k_{in}^{*}\right)$ ($\pi \left(k_{out}^{*}\right)$) is time-dependent, we use the rank function $R(\alpha ;k_{in}^{*},t)$ (resp., $R(\alpha ;k_{out}^{*},t)$). We compute this rank function for each link addition to a node with in-degree $k_{in}^{*}$ (resp., $k_{out}^{*}$) at each time *t*. Specifically:(3)$$\begin{array}{ccc} R(\alpha ;k^{*},t) & = & \frac{\sum_{k=0}^{k^{*}-1}n\left(k,t\right)\;k^{\alpha }}{\sum_{k}n\left(k,t\right)\;k^{\alpha }}. \end{array}$$

The sum in the denominator runs for all nodes whose in-degree is lower than $k_{in}^{*}$ (resp., out-degree is lower than $k_{out}^{*}$). When a new edge is created, the target (resp., the source) node is drawn with a probability for a given $\alpha_{o}^{in}$ (resp., $\alpha_{o}^{out}$). We insert this α into Equation (Equation 3).

We choose the value $\alpha_{o}$ that minimizes the corresponding Kolmogorov–Smirnov (K-S) distance between the empirical cumulative distribution function (ECDF) and the theoretical linear CDF distribution. We sample 10% of all the edges in the transaction network and calculate K–S distance between the empirical distribution and a theoretical one, in this case a power law, for a range of α∈[0, 2.5].

### 2.5. Deep Dive Methodology

We devise the following methodology for our deep dive study:We extract from a fully synchronised archive Ethereum node all transactions from the Ethereum blockchain involving transfers of tokens.We build the corresponding directed transaction networks for four DeFi-relevant tokens: Ampleforth (AMP), Basic Attention Token (BAT), DAI and Uniswap.We calculate the correlation between in- and out-degree distributions both for all nodes in the network and for high-degree nodes to see whether high-degree nodes are more correlated.We compare in- and out-degree distributions with potential best fit functions for the degree distributions.We study the centralisation of the network, analysing the preferential attachment to identify the role of hubs. We use the K–S distance between the empirical degree distributions and a power law degree distribution in the range [0, 2.5] for all tokens using the rank function to identify the K–S distance between the empirical cumulative distribution function (ECDF) and the theoretical CDF function.We explore the evolution throughout time of the of the value of α in the rank function to confirm the super-linear attachment in the transaction networks.We calculate how the density, i.e., the number of edges, grows with the growth in size of the network to confirm our preferential attachment study.We study network dismantling by removing up to 200 addresses (represented as nodes in the network) with the highest degree, and we observe how the network is impacted via the Largest Strongly Connected Component (LSCC) over network size ratio. For this, we use three strategies based on the type of node to remove.We study the evolution of the scalar assortativity of the transaction networks during dismantling.

### 2.6. Multilayer Network-Based Methodology

Multilayer networks consist of networks where nodes share not only one type of edge describing one relation type, but multiple types of edges describing different relations. The multilayer network approach has been proficiently used in many different fields, from transportation [31] to veterinary and livestock epidemics [32]. In this domain, we find a set of concepts and methodologies such as multilayer networks, temporal networks, multiplex networks, interdependent networks and networks of networks. The seminal works of De Domenico et al. [33] in 2013 and Kivela et al. [34] in 2014 contributed to clarify multilayer networks.

The Ethereum network is multilayered with respect to its ERC-20 tokens: addresses sending or receiving native Ether may appear not only in the Ether transaction network but also in one or multiple token transaction networks. We study a large number of tokens through a multilayer network approach to assess structural similarities between token networks.

For that, we build a multilayer network based on the overlap between addresses in Ethereum and ERC-20 token networks. Only addresses that participate in both non-zero Ether and token transfers are included. From the original Ether transaction network *E* and the original token network *T*, we create two new networks $E^{\prime }$ and $T^{\prime }$ including only addresses *a* where a∈E∩T, and the edges *e* that connect them, respectively, in *E* and *T*. $E^{\prime }$ and $T^{\prime }$ are therefore two layers, also called two aspects, in a multilayer network. To avoid too small or inconsistent networks, we limit our analysis to the transaction networks sharing at least 10,000 addresses with the Ethereum network, up to the block 12,500,000 (24 May 2021). In this way, we obtain 1768 Ether-token multilayer networks.

To make our computations easier, we transform the directed networks into undirected networks. The move from directed to undirected networks implies the move from the LSCC parameter to the Largest Connected Component (LCC) parameter. We dismantle these networks in an iterative way using the top 100 high-degree nodes: this threshold—that our deep dive proves to be significant—was also used in Section 3.4.

After the creation of these multilayer networks, our methodology consists of the following steps:We measure the initial size of the resulting networks in terms of number of nodes and plot the histograms of its distribution.We calculate the distribution of the initial LCC over network size ratio for $T^{\prime }$ and $E^{\prime }$ to characterize the different networks.We perform a similar exercise in each layer after 100 highest degree iterative node removals in the other layer to observe how dismantling takes place in each network.We explore the relation between the LCC over network size ratio reached after dismantling in the Ether and the token network.We study the relation between the degrees of a node in different layers.

### 2.7. Bird’s Eye Methodology

We study full Ether and ERC-20 token networks up to block 12,500,000 (24 May 2021). We focus on the top 100 highest degree addresses obtained through iterative dismantling and analyse them to assess structural similarities. Although arbitrary, this threshold of 100 proved to be both significant and computationally feasible in Section 2.6 and Section 3.4. As in Section 2.6, we only compare token networks consisting of at least 10k nodes (addresses) to skip too small networks which might lead us to inconclusive results. In this case, we obtain 2432 networks, in contrast with the 1768 analysed in Section 2.6. Our research objective is to study the similarity of these top 100 nodes across different networks.

We use the Jaccard Index as a first measure of this similarity per set. However, the Jaccard Index $J_{A,B}$ compares the similarities between two sets *A* and *B* regardless of the ordering of its elements in the following way:(4)$$J_{A,B}=\frac{A\cap B}{A\cup B}.$$

Other existing similarity measures are also not well suited for comparing the rankings of elements where absolute, as opposed to relative, ordering is relevant. In our case, this is important because our aim is to weight similarity between hubs more strongly than similarity between lower ranking nodes. We considered as potential alternatives several string similarity metrics which can be adapted to compare similarity between two lists of addresses. The Levenshtein distance is a string metric that measures the difference between two sequences of characters. In our case, it can be used to compare two lists of addresses. However, it takes into consideration relative instead of absolute distance, while in our case, we give more weight to the comparison of highly ranked addresses. In a similar fashion, the Hamming distance measures the minimum number of substitutions required to change one sequence into a different one with the same length. Again, there is no weight depending on the absolute position of edits, so similarity in the lower ranking addresses can overpower similarity in higher ranking ones, which deviates from our objective. A similar case happens with most of the other edit distance measures, such as the Jaro–Winckler distance: they are only suitable to compare strings. Finally, we also considered the Kendall Tau correlation rank, but it uses relative as opposed to absolute positions to evaluate concordant and discordant rankings between elements. We need a measure that considers the absolute ordering or unique elements in its definition to identify the likelihood of a node to be a hub.

In Section 3.7 we provide a definition of the Ordered Jaccard Index which addresses the requirements mentioned above. This new index weights a set of elements based on their absolute order. This way, similarity between elements in higher ranking positions outweighs similarity between elements in lower ranking positions. This proposed index gives more relevance to the ranking of highly connected nodes (hubs).

These are the steps that we follow for our bird’s eye similarity methodology:We plot the Jaccard Index between every token network *T* and the one of Ether *E*. We do this pairwise for every one of our tokens.We analyse the in-degrees of the 2432 token networks using the model selection for power law fits proposed by Alstott et al. [35]) in 2014.We propose a novel similarity measure that is order-sensitive and potentially useful for our case of networks with very different sizes, and we test its meaningfulness using a set of randomized null models.We draw a heat map with this new similarity measure between all analysed tokens.We plot the histograms of the distributions of both the Jaccard Index and our new similarity metric, called Ordered Jaccard, measured between every individual token and Ether, and we calculate their correlation.We research potential similarity drivers: network size and market capitalisation. We calculate their correlation.We plot network size and the Jaccard and Ordered Jaccard Indexes between the Ether and token networks to identify any potential correlation. Here we start to detect a pattern.We perform a similar exercise with the maximum market capitalisation reached by every token network.We study the relation between Ether and a specific token and that token with all other Ethereum-based tokens via the Ordered Jaccard Index.We explain the methods that we follow to identify addresses belonging to exchanges and smart contracts.We plot the percentage of exchanges and smart contracts in the top 100 high-degree nodes.Finally, we plot a histogram with the reappearance frequency of dismantled nodes across networks.

## 3. Results

### 3.1. In- and Out-Degree Correlation

For our deep dive on four tokens, AMP, BAT, DAI and UNI, we first calculate in Table 2 the correlations between in- and out-degrees for all nodes and for nodes specifically with degrees over 100. We observe that correlation is usually stronger for these higher degree nodes. Above a certain threshold, we are more likely to have many in-going and out-going transactions.

### 3.2. Power Law Fit

Second, we obtain the potential best fits for the in- and out-degree distributions for the four tokens AMP, BAT, DAI and UNI. Table 3 shows the potential power law fits obtained. Figure 1 shows the CCDF’s of the four tokens [36].

### 3.3. Preferential Attachment

In the four tokens analysed, we consistently observe that the minimum value of α (see Equation (Equation 3)) is achieved around 1.0 for the out-degree and around 1.1 for the in-degree. A value of α>1 for the in-degree indicates a super-linear preferential attachment, with the consequence that a small number of nodes attracts most of the connections in the network and will eventually form super-hubs. This is another indication of the rising centralisation in the network, possibly caused by the presence of key smart contract and exchange addresses. Figure 2 shows the K-S distance values for the whole range of $\alpha^{in}$ and $\alpha^{out}$ for all four tokens. The minimum values of K-S distance indicate evidence of super-linear preferential attachment in all four tokens as Table 4 shows. This preferential attachment is an indication of an increasing centralisation in the transaction networks. Our hypothesis is that smart contract nodes and exchanges effectively act as hubs in the transaction network.

We also explore the evolution of α through time with disjoint and non-cumulative time windows. The size of the transaction networks makes this task computationally demanding, so we only analyse a random 10% sample of the new edges [14]. We repeat this process multiple times and calculate the average value of α for each window. Figure 3 shows the final results.

Additionally, Figure 4 plots network density as a function of network size for the four tokens. We observe that network density is inversely proportional to network size $d\propto N^{-1}$. This means that the number of edges grows linearly with the size of the network: transactions mostly reuse already existing edges. New nodes joining the network only add a limited number of new edges. This is consistent with our observation regarding preferential attachment.

### 3.4. Network Dismantling

Network dismantling studies how to find a minimal number of nodes whose removal dismantles a network [30] into isolated sub-components. It belongs to a class of nondeterministic polynomial hard (NP-hard) problems. This means that there is no efficient algorithm that can find the optimal set of nodes to remove for large-scale networks. However, there are approximate methods which work sufficiently well in practice even for large networks [28,29]. This analysis focuses on estimating the influence that the different types of nodes have on dismantling and not on finding the most efficient dismantling strategy. We compare addresses of smart contracts, nodes controlled by code, and known exchanges, with nodes corresponding to the addresses of the externally owned accounts (EOA), controlled by the actual users possessing the corresponding cryptographic keys.

We perform dismantling by repeatedly removing nodes of the appropriate type with the highest in-degree $k_{in}$ one by one and recalculating the in-degree for all remaining nodes after each removal. As a measure of dismantling, we use the ratio of the Largest Strongly Connected Component (LSCC) to the total network size. LSCC is defined by the largest set of graph nodes such that, for every pair of nodes *a* and *b*, there is a directed path both from *a* to *b* and from *b* to *a*. In our analysis, we perform dismantling for up to 200 nodes of each type and for the four tokens separately as shown in Figure 5. We observe that for all of them the removal of nodes corresponding to the addresses of smart contracts and known exchanges causes faster dismantling than the removal of nodes corresponding only to the addresses of EOAs. By removing just a handful of nodes corresponding to the addresses of smart contracts and known exchanges, the LSCC collapses, and a transition phase in the network occurs. This indicates a large structural centralisation where those removed nodes were effectively acting as hubs in the transaction network, involved in the majority of the transactions. In the information security realm, intentional risk managers should protect these nodes the most [14,15,39,40,41]. We also performed additional dismantling for up to 10k nodes for each of the tokens, but this did not show qualitatively different results. So in Figure 5, we only show results for up to 200 nodes. We confirm that the dismantling threshold of 200 nodes was even too large: the removal of first 100 nodes seems sufficient for our analysis.

### 3.5. Assortativity

The assortativity coefficient *r* measures whether nodes of a certain degree $k_{i}$ tend to form a connection with other nodes with similar degree. Its range is −1<r<1. A positive value indicates assortative mixing. This means a high correlation between the degrees of neighboring nodes which usually indicates the presence of communities: nodes tend to link with nodes with a similar degree. A value close to zero suggests non-assortative mixing. This means that the degree correlation between nodes is very low: nodes tend to link with other nodes regardless of their degree. Finally, a negative value reveals disassortative mixing: nodes tend to link with nodes with a very different degree. Equation (Equation 5) presents the standard definition of the assortativity coefficient *r* [42], where $a_{i}=\sum_{j}e_{ij}$, $b_{j}=\sum_{i}e_{ij}$ and $e_{ij}$ is a fraction of edges from nodes of degree $k_{i}$ to nodes of degree $k_{j}$.
(5)$$r=\frac{\sum_{i}e_{ii}-\sum_{i}a_{i}b_{i}}{\sigma_{a}\sigma_{b}}$$

In Figure 6, we show the assortativity of the transaction networks during dismantling for two types of nodes, nodes corresponding to the addresses of smart contracts and known exchanges (blue line) and nodes corresponding to EOA addresses (orange line). As reference, a green line for highest in-degree dismantling without considering the type of address is provided. Since for this deep dive we are analyzing directed graphs, we will here consider in-degree assortativity. The initial assortativity of networks is slightly negative but close to 0 (from −0.09 for BAT and UNI to −0.21 for AMP).This is not surprising considering the existence of centralisation in the network: most of the low nodes are connected to the large central hubs with very few connections between them. Removal of nodes corresponding to EOA addresses during dismantling has no discerning effect on the assortativity, while for contracts and known exchanges, the assortativity tends to increase towards zero, making the networks less centralised and almost non-assortative. This is probably because the first nodes to be removed during dismantling are the highly connected hubs. The removal of these nodes first raises the assortativity in the network.

### 3.6. Multilayer Network Dismantling

The results in the previous sections, inspired by our previous work on transaction networks of DeFi-collateral Ethereum-Based tokens [26], reveal the structural role of smart contracts and known exchanges in the transaction networks of these four tokens. The rest of this article focuses on whether we can generalize these results for other tokens in Ethereum. We study a larger number of tokens through a multilayer network approach to assess structural similarities between token networks. For that, we expand our focus of attention, and together with ERC-20 token transfers, we also consider the native token, Ether, and its transaction network.

Our aim is to understand what happens to aspect $E^{\prime }$ of Ether when we dismantle, i.e., remove nodes in aspect $T^{\prime }$, the token network within the multilayer network, and vice versa. We follow a “greedy” strategy to dismantle networks by removing high-degree addresses from another layer of the multilayer network. Our hypothesis is that pairs of tokens whose transaction networks are efficiently dismantled by this approach have a similar structure.

We measure the initial size of the resulting network and plot the histograms of its distribution in Figure 7. By construction, the two derived networks, $E^{\prime }$ and $T^{\prime }$, have the same size, since the addresses need to appear in both *E* and *T*. The majority of the networks have less than 100,000 nodes, with the vast majority having around 10,000 nodes.

This network size is much smaller than the original one of the complete *E* network, with 125,472,929 nodes. Only non-zero Ether transactions appear in $E^{\prime }$. Zero-value Ether transactions, which might occur in tokens, are not included since to transfer value they invoke methods of smart contracts, and they do not always have to pay transaction fees to the miner for that. Consequently, the overlap between $E^{\prime }$ and $T^{\prime }$ edges is small: these two transaction networks have identical nodes (addresses) but different edges as transactions between these addresses are not the same, i.e., one aspect or member network includes only Ether transfers, and the other includes token transfers.

Figure 8 shows the distribution of the initial LCC over network size ratio for $T^{\prime }$ and $E^{\prime }$. The LCCs in $T^{\prime }$ cover their network more densely (top left) than the aspect $E^{\prime }$ (top right), which is much smaller than the original network *E*. $T^{\prime }$, however, is more similar to the original token network *T*.

We also depict two similar graphs after 100 removals using the Δ between $LCC_{0}/N$ at the initial stage and $LCC_{100}/N$. A value closer to zero shows negligible dismantling effects, while a value close to one indicates a complete dismantling. As we deal with multilayer networks, we identify the high-degree nodes to remove in one layer (aspect), and we remove them in the other layer (aspect). Identifying the high-degree nodes and removing them in the same layer would have unsurprisingly “demolished” the LCCs, as Section 3.4 showed. In this case, dismantling in the Ethereum layer first (bottom left) has a weaker effect that dismantling in the token layer first (bottom right) on these multilayer networks.

We explore whether there is a relation between the LCC over network size ratio in the Ether network and the LCC over network size ratio in the token network and between their respective Δs. We plot these two values together in Figure 9. We obtain a Pearson ρ correlation of −0.17 between the initial LCC over network size ratio for token and Ether networks and a correlation between the Δ of the LCC over network size ratio for token and Ether networks using the Ether-first and token-first dismantling of 0.32. This path leads to no results.

We then plot the histogram of the distribution of degree correlation across the same nodes in networks $E^{\prime }$ and $T^{\prime }$ to understand how the degree of the same node correlates with its degree in the other network. The histogram of inter-network degree correlation, as in Figure 10 even though skewed to zero, is fairly stretched up to one, which is compatible with our hypothesis mentioned in Section 3.6.

We find a significant relation between degrees of nodes present in the two layers (aspects) of the multilayer network. This finding helps identify target nodes to dismantle. Figure 11 supports this point. We calculate a Pearson correlation between degrees in correlation in $T^{\prime }$ and $E^{\prime }$ of 0.53 and 0.51, respectively.

Unfortunately, the size of token networks is usually too small to be compared to the Ether network. With this approach, we sample a significant portion of the token network but not enough of the much larger Ether network. This size gap complicates the possibility to infer properties for a token layer based on the Ether layer and vice versa. We have to study network similarities in a different way to obtain meaningful results.

### 3.7. Bird’s Eye View of Ethereum-Based Tokens

As explained in the methodological section, we now consider the complete Ether network and the full token transaction networks. We investigates the 2432 tokens that have more than 10,000 transactions overall. We build an undirected cumulative transaction network, and through the approach of iterative network dismantling, we obtain a set of their most significant nodes. What can we say about these sets of nodes? Do they look similar between Ether and the individual tokens?

As the top plot in Figure 12 shows, the maximum Jaccard Index between them is about 0.08. In our scenario, this means that 8 addresses of the top 100 high-degree addresses are shared between the Ether and the token networks. The Jaccard Index presents a major drawback for our study: it does not consider the position in the degree distribution of the addresses used for its calculation. However, ordering of nodes and their degrees matter in our analysis. An example of this importance is the result that we obtain in Section 3.3 when we study preferential attachment in AMP, DAI, BAT and UNI. Other studies [14,15,40] mention as well a heavy tail in the degree distribution for other tokens.

Victor et al. [12] in 2019 indicated that only a small fraction of the degree distributions of the transaction networks in ERC-20 tokens follow a power law: in many of them, very few nodes have a very high degree. Our analysis of the in-degree distribution of 2432 token networks, following the method of Alstott et al. [35], shows that 1693 networks are better approximated by a power law, 610 by a truncated power law, 88 by a lognormal, 20 by a lognormal positive, 18 by an exponential and just 3 by a stretched exponential. We considered a better fit only when their p-value of distribution comparison was acceptable (lower than 0.05). Fitted $x_{min}$ show a mean of 20.2 and a standard deviation of 95.2, mostly driven by very few outliers. The median is four and the 95% percentile is still at nine. However, just a high *p*-value is not sufficient to state that a token follows a power-law or truncated power-law degree distribution. The γ exponent shows a mean of 2.4 and a standard deviation of 0.87: here the median is 2.14, and the 75% percentile is 2.84. Clauset et al. [43] confirm how rare power law functions in reality are. In any case, the position of the node representing an address in the ordered degree distribution is valuable information that the pure Jaccard Index does not consider.

Our interest transcends the number of nodes that overlaps between two networks. The absolute ordering of those nodes in each of the networks is also important. If a particular address appears in two different networks with the same ranking, for example within the top 10 high-degree nodes, this indicates that this node, as a hub, probably plays a similar structural role in both networks.

As the Jaccard Index *J* returns a value regardless of the node degree rank, we re-define a Jaccard-like measure as an order-sensitive measure that might provide information even for networks with very different sizes. The order we introduce in the set is based on the degree.

We define an Ordered Jaccard Index $O_{P,Q}^{j}$ between two ordered sets *P* and *Q* in the following way:(6)$$O_{P,Q}^{j}=\left\{\begin{array}{cc} 0 & P\cap Q=\emptyset \\ \frac{\sum_{\forall a\in (P\cap Q)}\frac{2}{Pos_{P}\left(a\right)+Pos_{Q}\left(a\right)}}{\sum_{\forall b\in P}\frac{1}{Pos_{P}\left(b\right)}} & P\cap Q\neq \emptyset ,|P|=|Q| \end{array}\right.$$
where $Pos_{P}\left(a\right)$ is a position of element *a* from ordered set *P* ranging from 1 to |P|.

With this proposed measure, we weigh a set of elements based on their order and independently of the size of the respective original network. This way, having the same element in first position is more important than having the same element in second or *n* position, by a contribution weighted as the inverse of the position the elements occupy. If they are in a different position, an average of the two positions is taken, regardless of whether the resulting weight exists as a real position in the discrete ordering set. If the elements are not present in both sets, they contribute 0 to the similarity of the ordered set. We normalize the sum of the individual contributions by the sum of the harmonic series 1/n for all the positions overall present in the set. In the formula, we take one of the sets in the denominator, given that the sets have the same size. For ordered sets of different sizes, they can easily be capped to their minimal common size.

The distribution of the values that we obtain, as mid and bottom plots show in Figure 12, is more spread than the one of the conventional Jaccard Index *J*.

We test the meaningfulness of this, compared to the Jaccard Index it has a more stretched distribution against a randomized null model. From the 161,000 nodes effectively appearing in the top 100 degree distributions, we extract the frequency for each of the nodes, i.e., their probability to occur. From this distribution, we take 2432 random buckets of 100 samples plus another one simulating the Ether sample. We repeat this experiment 10 times. Then, we plot this null model against the real data with a Cumulative Distribution Function (CDF) plot in Figure 13.

Our proposed Ordered Jaccard Index brings an advantage: if we plot the matrix of the Jaccard Index between all 2432 tokens with more than 10,000 nodes at the time of data extraction, we do not see any visible correlation pattern. We know that we have addresses in common but not how important they are. However, if we draw the matrix of the Ordered Jaccard Index $O_{j}$, we start detecting some interesting patterns as the lower right corner of Figure 14 displays.

The token networks that we observe differ not only in their size, but in their construct, scope and purpose. For example, in our deep dive presented in Section 2.1, we studied an algorithmic stablecoin, a utility token, a multi-currency pegged stablecoin and a governance token. There are even more, in a way that makes finding the similarities between them not trivial.

Figure 12 plots the histograms of the distributions of the Jaccard and the Ordered Jaccard Indexes measured between every individual token (we count 2432 of them) and Ether. We find an expected high correlation, around 0.91 between the Ordered Jaccard Index and the normal Jaccard Index. About 1000 of the investigated samples do not share any addresses, while the remaining display a different level of similarity, up to 0.08 (*J*) and 0.25 ($O_{j}$). The introduction of ordering in the proposed Ordered Jaccard Index makes this metric more sensitive to our study of the structural relevance of nodes when dismantling a network.

The next step is to research what could drive this similarity. First, we analyse the size *N* of the transaction networks that we have built. The bigger the size of the network is, the greater the “success” for a specific cryptoasset. Second, we collected market capitalisations from online cryptoasset market information aggregators such as CoinMarketCap [17] and CoinGecko [44]. These aggregators collect market data from a variety of popular exchanges covering a large collection of cryptocurrencies and tokens. We took the Ethereum network snapshot in May 2021, and we analysed market capitalisation data up to the same time span. If we consider a success in capitalisation as reaching (or having reached in the past) high market prices, Figure 15 reflects the correlation between the most successful tokens and those with the highest number of transactions and token transfers. We obtain a Pearson correlation ρ of 0.3318.

Figure 16 plots the network size *N*, i.e., the number of addresses taking part in the transaction network, and the Jaccard *J* (top left) and the Ordered Jaccard $O_{j}$ (top right) Indexes between the Ether and token networks (top plots). We start to see a correlation between network size and these two indexes. The two bottom plots show the maximum market capitalisation reached in aggregators such as CoinMarketCap [17] and CoinGecko [44], where we observe a stronger correlation. Our hypothesis is that market capitalisation, driven by interest from the investors, causes transfers and transactions, which increase network size, causing the most important exchanges to take an active part in the token transaction network thereby driving the similarity recorded by the Jaccard and the Ordered Jaccard Indexes between the hubs in different token networks against the native token Ether.

We reach the core of our study: the Ether network and the Ether-based token networks share addresses in the top degree list of nodes. They play an important role in holding up the network. This implies a structural similarity between tokens and Ether. We prove that the Ordered Jaccard Index between Ether and Ether-based tokens is correlated with the average of the Ordered Jaccard Index between the token and all other tokens with more than 10,000 addresses in their transaction network. Specifically, we measure the Ordered Jaccard Index with Ether and the average of the Ordered Jaccard Index with all the other tokens, and we find a strong correlation 0.674613, as Figure 17 depicts. Therefore, this measure can be deemed significant for similarities between tokens.

Market success and interest from investors seem to drive higher prices (market capitalisation). Higher prices drive activity in terms of transactions, i.e., token transfers; hence there are more addresses participating in the transaction networks, which raises the cumulative network size. Interest from investors causes tokens to be traded from addresses that are exchanges and/or smart contracts, i.e., the automatic engine of Decentralised Finance, DeFi.

There is no algorithmic way to know in advance which addresses are associated to exchanges. We resort to reports by *Chainanalysis* [45] and to the list of addresses provided by *etherscan.io* [46] listing exchanges. We combine the lists of addresses EOA (External Owned Account) and SC (smart contracts) related to exchanges, as both can be used for this purpose, the former mostly by centralised exchanges and the latter by decentralised exchanges. We use these lists to identify those addresses in the 100 high-degree addresses list for every token that is linked to exchanges, as Figure 18 shows.

We observe that exchanges are present in over 50% of the networks analysed, and they constitute up to 40% of the 100 top high-degree addresses in the network. For more than 50% of our analysed sample, they play a very important role. We assume that their real role might be even more relevant. While the information from *etherscan* is the best that we can obtain, it is not complete. Individual tokens might have very specific exchanges which are not yet listed or tagged.

We turn to a more “deterministic” view to identify addresses belonging to smart contracts because we can confirm SC addresses algorithmically. Figure 19 shows the results. They are comparable to exchanges: in about 50% of the analysed networks, we do not encounter addresses of smart contracts; however, the figures for those where they are present are much more skewed to the right. We have tokens in which smart contracts are up to 90% of the analysed set, while addresses related to exchanges were never more than 40%.

The results are consistent, even considering that we might be counting the smart contracts that are part of a known exchange in both histograms. In many token networks, a large set of the most important nodes, i.e., highly connected, correspond to smart contracts. Lastly, as we investigated similarities between high-degree nodes, Figure 20 plots a histogram of the number of token networks in which these nodes appear. The networks that we analyse still correspond to the 2432 token networks with more than 10,000 addresses, as described in Section 2.

Of the 243,200 (2432×100) possible different addresses that we could have had, our data show just 160098: so roughly 2/3 of addresses are unique. The majority of them appear only in one token network, but some others reappear in tens or hundreds of networks, such as EtherDelta2 Smart Contract (address: t0x8d12a197cb00d4747a1fe03395095ce2a5cc681) which occurs in more than 1000 networks, or IDEX 1 Exchange (address: 0x2a0c0dbecc7e4d658f48e01e3fa353f44050c208), a very popular decentralised exchange, or an exchange Hotbit (address: 0x274f3c32c90517975e29dfc209a23f315c1e5fc7) with 402 presences across different tokens and addresses. On the other hand, some of the most active addresses are hard to identify, with no official flags and labels. For example, we find a smart contract (address: 0x74de5d4fcbf63e00296fd95d33236b9794016631) with just 19 transactions in Ether but with more than 11M ERC-20 transactions at the time of writing. The very fact that such an important address could leave no recognizable trace of its identity, purpose or origin is a good testimony of the current challenges to the scientific and forensic analysis of public blockchains.

## 4. Discussion

The aim of Decentralised Finance (DeFi) is to provide a permissionless and secure alternative to the traditional financial system by using automated smart contracts that run on a decentralised public blockchain. As of April 2022, the total value locked within the DeFi ecosystem was well over USD 200B, although in May 2022 it decreased to USD 76B [47]. Although the blockchain protocol itself is decentralised, meaning that no single actor can easily influence the execution of transactions on a protocol level, there are indications of increased centralisation at the application level because of the reliance on smart contracts, which provide programming logic for automated financial services, and exchanges. To investigate whether and to what extent this centralisation happens, we analysed the transaction networks of different DeFi-collateral tokens and assessed the structural roles of two fundamentally different types of addresses: smart contracts (SC), controlled by the logic of the code inside the blockchain, and externally owned accounts (EOA), controlled by actual users or entities outside the blockchain. Exchanges can use both type of addresses, depending on their design (decentralised or centralised, automated or human-run).

Our analysis took three main research paths: the first one was inspired by our prior work on DeFi-collateral Ethereum-based tokens [26], in which we performed a deep dive, the second one experimented with multilayer networks, and the third one focused globally on the similarity between different token networks and the Ether network. The data that we analysed corresponds to the public blockchain of Ethereum from its creation up to block 12,500,000, created on 24 May 2021.

We observed an increasing centralisation in the transaction networks of all analysed tokens, with nodes corresponding to addresses of smart contracts and known exchanges acting as hubs.

We used several complementary methods to study Ethereum-based tokens. This was investigated following several methods:First, in our deep dive, we observed a slightly super-linear preferential attachment coefficient ($\alpha^{in}>1.0$), that is persistent throughout time. This implies that few nodes attract a majority of connections from new nodes. This resembles a form of “winner takes all” effect, commonly observed in social systems as well [48]. We identified the relevance of smart contracts and exchanges when we dismantled the resulting transaction networks following selective strategies with a special focus on SC and exchanges.Second, we studied a larger set of tokens and focused only on non-zero Ether and token transactions in networks that share at least 10k addresses with the Ether network. We used a multilayer network approach and tried to dismantle a specific layer (aspect) based on selective node-related information coming from a different layer (aspect). Although we confirmed again the relevance hubs when dismantling the layers, we abandoned this research path due to the big size difference between token networks and the Ether transaction network.Third, we broadened our lens and study similarities in all transaction networks with at least 10k addresses. For this, we came up with a new index, the Ordered Jaccard Index, that facilitated and confirmed our findings regarding the structural role of SC and exchanges in these networks. We completed our analysis by identifying a degree of correlation between this new index and network size and even market capitalisation.

The Jaccard Index has proven very useful in many problems ranging from information theory to machine learning [49]. Our Ordered Jaccard Index can be interpreted as a specific application of the weighted Jaccard Index [50] or generalized Jaccard Index where we have an array of weights for the elements corresponding to their ordering. If we consider that the weighted Jaccard Index satisfies triangle inequality [51], we can assume as well that our Ordered Jaccard Index measure satisfies triangle inequality in which positions effectively act as weights. We suggest to prove this in future analyses.

The evidence for increased centralisation in the application layer of Ethereum, currently the most popular smart contract public blockchain platform, should not come as a surprise. We argue that smart contracts effectively centralise the application logic within the decentralised application (DApp) ecosystem, including the decentralised finance (DeFi) ecosystem in particular.

This centralisation effect is compounded by the existence of exchanges, which act as intermediaries for a majority of transactions, regardless of whether they are decentralised (with services being totally or partially implemented through smart contracts) or centralised (where most transactions are processed off-chain). Instead of a truly decentralised finance (DeFi) ecosystem, we observe a very centralised system where end users lack interest to interact with each other and rather rely on a limited number of smart contracts and exchanges to participate in most transactions.

Despite the initial promises of democratization and decentralisation of the crypto scene, its “automation infrastructure” of smart contracts and exchanges indeed plays a crucial role in holding together the transaction networks. Without this component, the networks would quickly collapse.

The current situation resembles the late 1990s and early 2000s when large Web service providers, online retailers, social media, news and messaging intermediaries, effectively centralised application logic and information flow within their respective service ecosystems in the Internet, notwithstanding the underlying Web technology, implemented on a completely decentralised Internet Protocol (IP). The results presented in this paper indicate that the distributed applications (DApp) ecosystem is, for the time being, on a similar centralisation trajectory, although the long-term consequences of this are yet to be fully described and experienced.

Future research directions will encompass the study of tokens running on different blockchains. Projects that have corresponding counterparts on different blockchains, such as Polygon and second-layer solutions, such as USDC and USDT stablecoins, are an attractive research area to analyse how their structures influence decentralisation at protocol and application levels. Additionally, a similar analysis could be performed on recent Proof-of-Stake blockchains that are increasingly popular, such as Polkadot, Solana and Cardano, many of which are developing cross-chain compatibility layers with Ethereum as well.

## Figures and Tables

**Figure 1 entropy-24-01048-f001:**
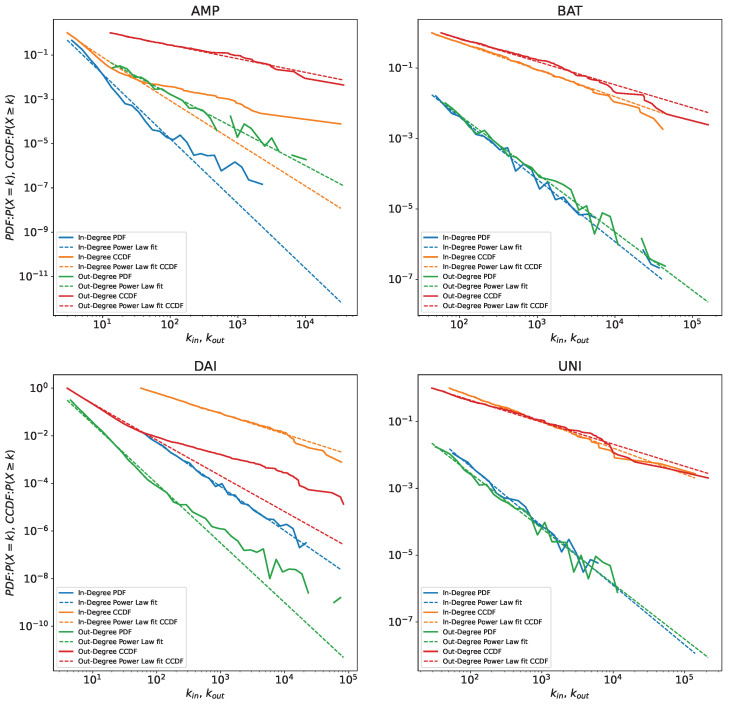
Empirical node in-degree and out-degree PDF (Probability Density Function) and CCDF (Complementary Cumulative Distribution Function) for each token plotted, together with the fitted values. To provide a better comparison, we always use fitted power laws as reference, even when they are not the best model as in Table 3. We notice the outliers in DAI out-degree distribution that make the best fit deviate from the power law and the distance between fitted CCDF and empirical CCDF for in-degrees of AMP, which might be caused by the smaller size of this network compared to the others. These are the cases where we encounter the smallest $x_{min}$ for our fits as well.

**Figure 2 entropy-24-01048-f002:**
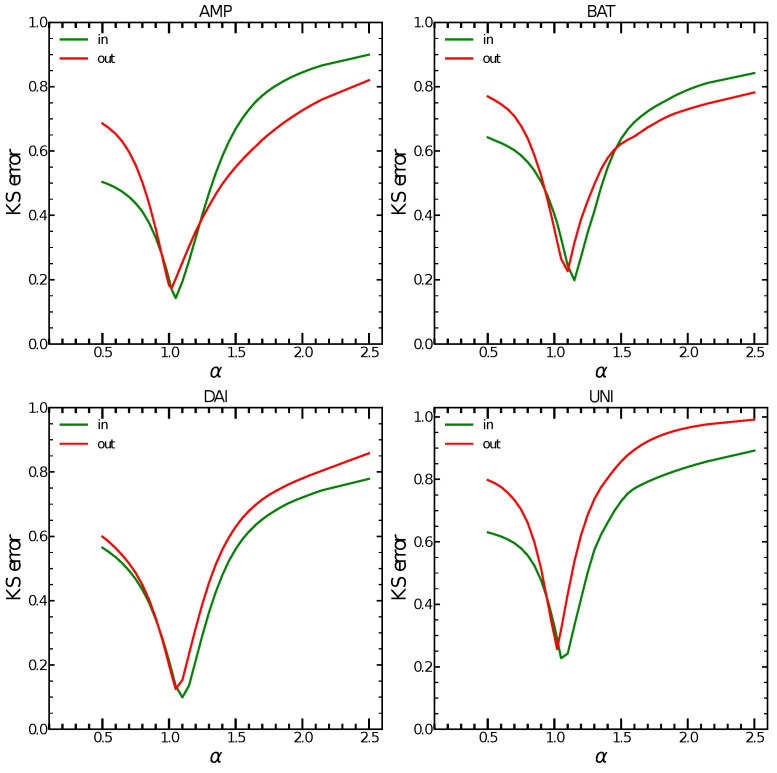
K-S distance between the empirical degree distribution and a power law degree distribution for the preferential attachment coefficients $\alpha^{in}$ and $\alpha^{out}$ in the range [0, 2.5] for all four tokens. Minimum values for all four tokens are achieved around 1.0 for the out-degree and around 1.1 for the in-degree which indicates super-linear preferential attachment.

**Figure 3 entropy-24-01048-f003:**
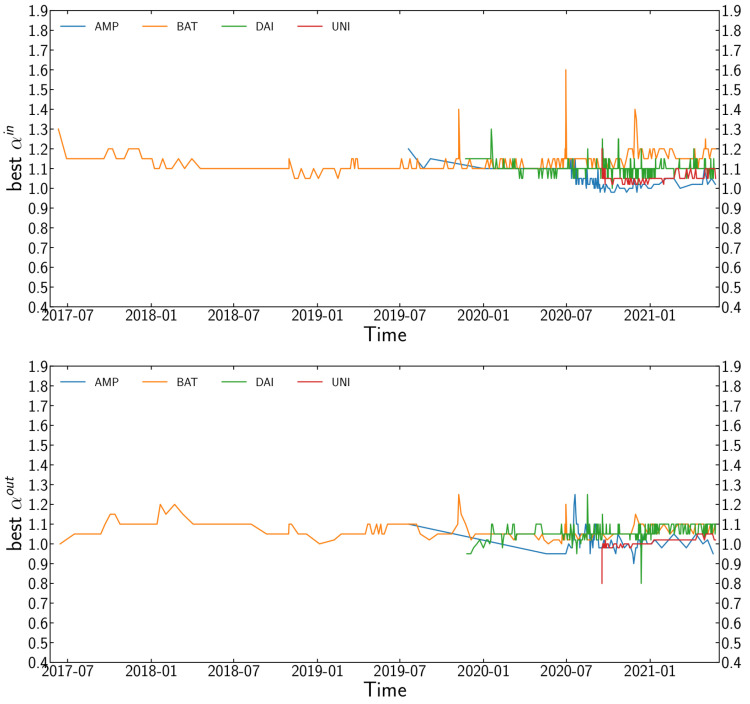
Evolution of the best fit for the in-degree $\alpha^{in}$ (**top** panel) and out-degree $\alpha^{out}$ (**bottom** panel) preferential attachment coefficients for all four tokens. We used disjoint and non-cumulative time windows to define sub-networks on which we calculate the preferential attachment. Both $\alpha^{in}$ and $\alpha^{out}$ are almost always above 1.0 throughout the time period studied, which indicates the persistence of the super-linear preferential attachment.

**Figure 4 entropy-24-01048-f004:**
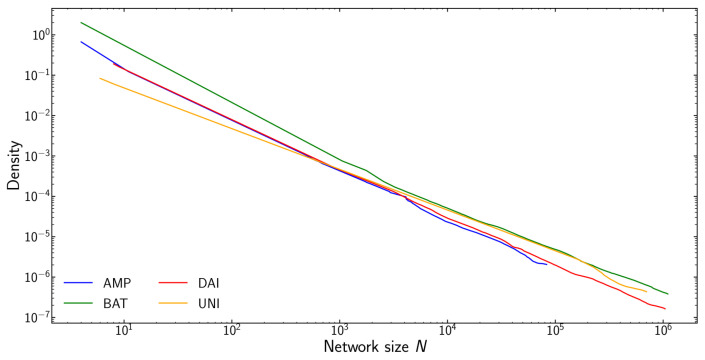
Network density as a function of network size *N*. The addition of new nodes to the network does not densify the network. The number of edges scales as $d\propto N^{-1}$. Each new node adds only a limited number of new edges. This is in line with the observed preferential attachment.

**Figure 5 entropy-24-01048-f005:**
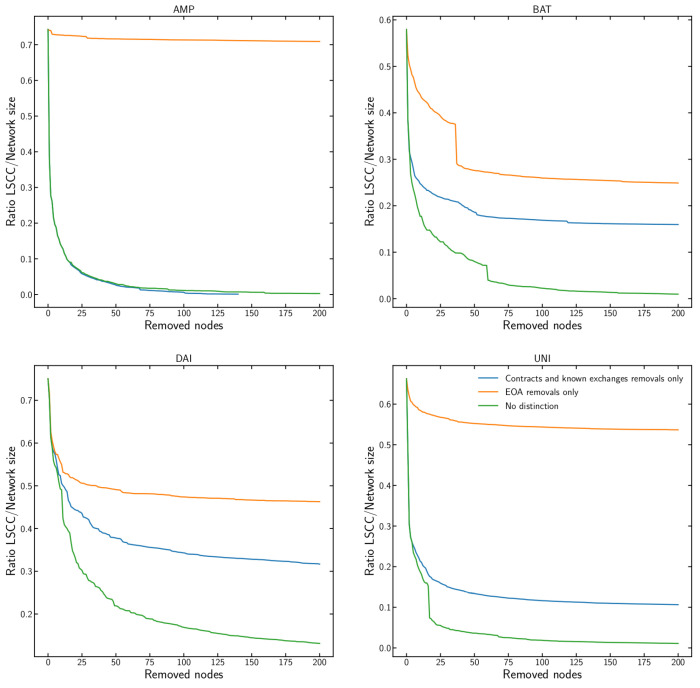
Dismantling of the Largest Strongly Connected Component (LSCC) in the transaction networks of the four tokens based on removing the nodes with the highest in-degree. We follow three strategies: first, we only remove nodes that correspond to smart contracts and known exchanges addresses; second, we remove nodes corresponding to EOA addresses; and third, we iteratively remove the highest degree nodes regardless of their address type (a “greedy” strategy, included here as benchmark). The removal of the nodes corresponding to the smart contract and known exchange addresses causes the fastest dismantling compared to EOA. This highlights the important structural role played by these addresses in the network.

**Figure 6 entropy-24-01048-f006:**
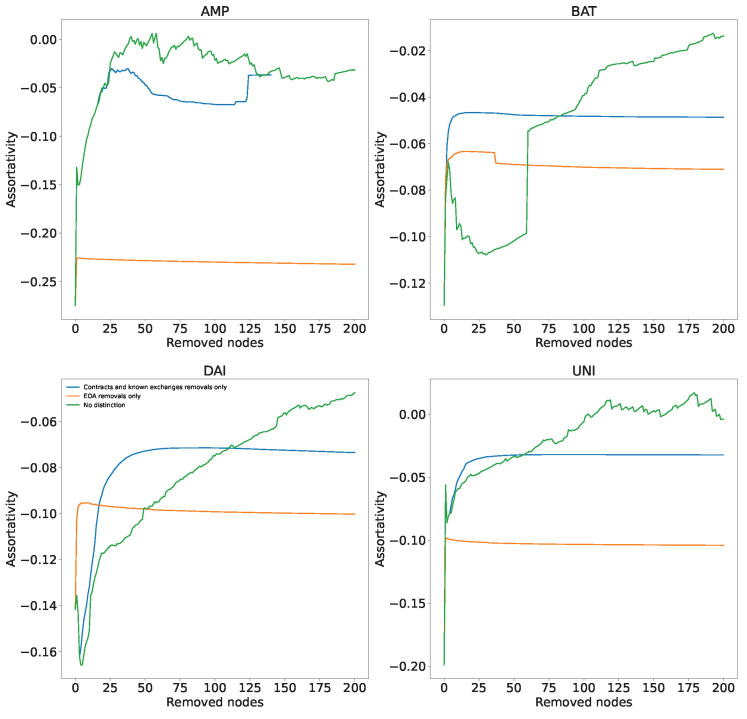
Evolution of the assortativity for in-degree of the transaction networks during dismantling. We follow the three strategies described in Figure 5. Initial assortativity in the four tokens is slightly negative, probably because most of the low in-degree nodes connect to large hubs and have few connections between themselves. This is why removing EOA nodes during the dismantling, which tend to have lower degree than smart contracts and exchanges, does not affect assortativity at all. However, the removal of smart contracts and known exchange nodes increases assortativity. It makes the network almost non-assortative, as many connections to these high-degree central nodes are removed.

**Figure 7 entropy-24-01048-f007:**
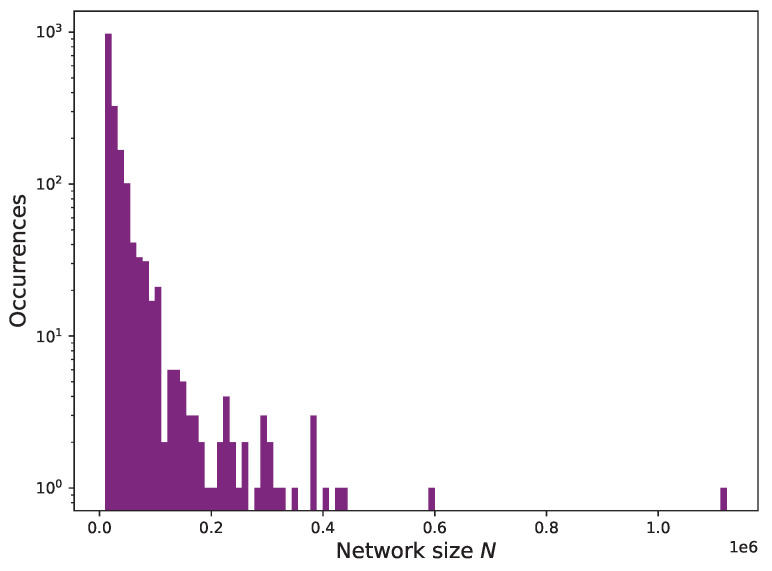
Size of the network layers $E^{\prime }$ and $T^{\prime }$ participating in the multilayer networks in terms of number of nodes (addresses) and their number of occurrences for each of the tokens considered.

**Figure 8 entropy-24-01048-f008:**
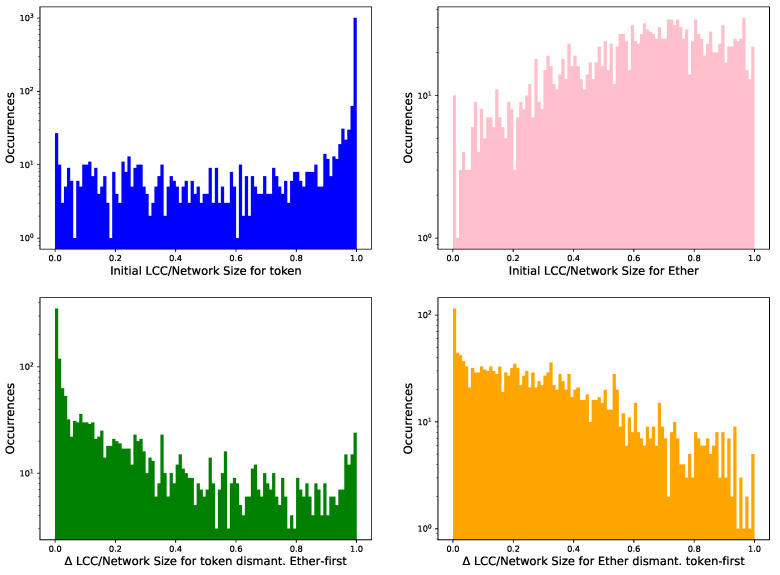
Histograms of the initial LCC over network size ratio for $T^{\prime }$ and $E^{\prime }$. In the panels below, we analyse the difference Δ, $LCC_{0}/N$ − $LCC_{100}/N$ to identify how much this ratio changes after dismantling one hundred nodes (of $T^{\prime }$ and $E^{\prime }$) with Ether-first (**bottom left**) and token-first (**bottom right**) strategies, respectively.

**Figure 9 entropy-24-01048-f009:**
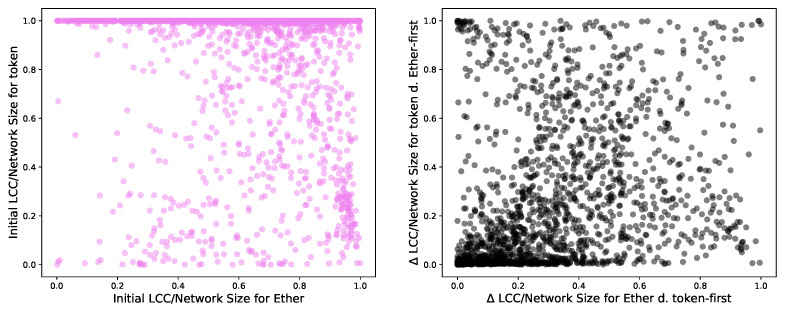
**Left**: LCC over initial network size ratio plotted for $E^{\prime }$ and $T^{\prime }$ computed for multilayer networks built with shared addresses between $E^{\prime }$ and $T^{\prime }$. We calculate a Pearson correlation of −0.17. **Right**: Plot of the Δs of LCC over network size for token and Ether networks reached after using the Ether-first and token-first dismantling. We obtain a correlation of 0.32.

**Figure 10 entropy-24-01048-f010:**
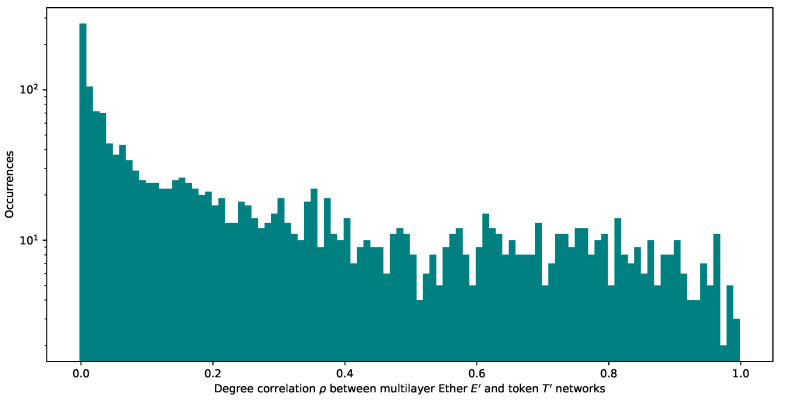
The degree correlation between the two multilayer networks $E^{\prime }$ for Ether and $T^{\prime }$ for token consisting only of addresses a∈E∩T.

**Figure 11 entropy-24-01048-f011:**
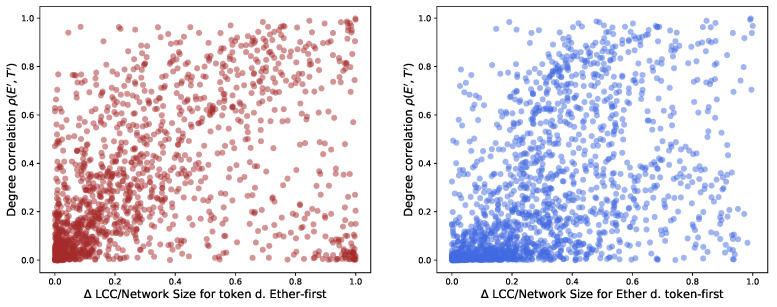
Degree correlation ρ between the two multilayer networks $E^{\prime }$ and $T^{\prime }$ plotted against the Δ of the LCC over network size ratio. This Δ refers to the portion of the LCC over network size ratio dismantled in $E^{\prime }$ and $T^{\prime }$. The Pearson correlation between $\rho (E^{\prime },T^{\prime })$ is, respectively, 0.532750 and 0.505823.

**Figure 12 entropy-24-01048-f012:**
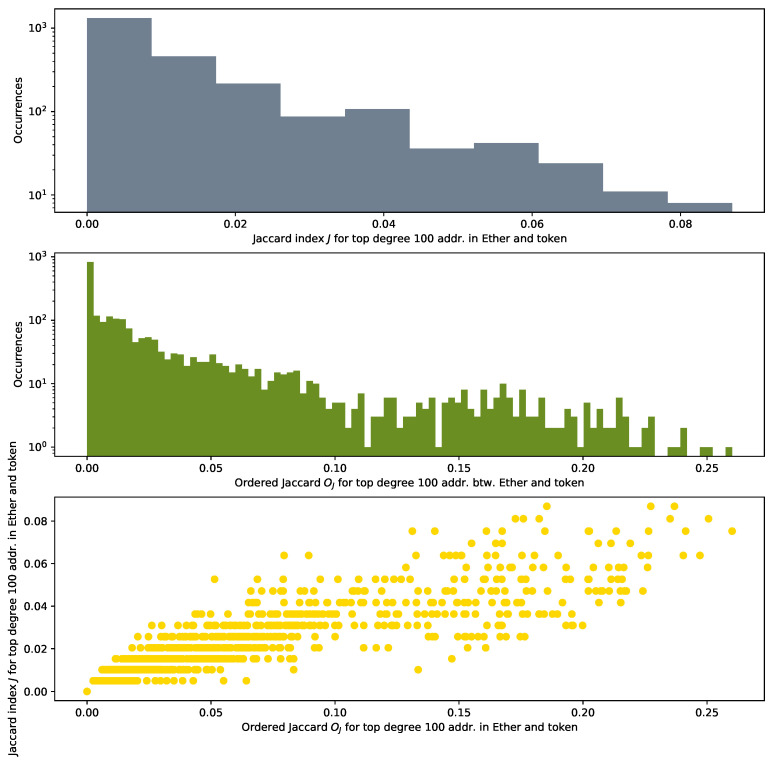
Histograms of the distribution of Jaccard Index *J* between the top 100 high-degree nodes (“greedy” dismantling order) for Ether and analysed tokens (**top** plot); plot of the Ordered Jaccard Index $O_{j}$ (**mid** plot); plot of $O_{j}$ and *J* (**bottom** plot); the correlation between $O_{j}$ and *J* is, as expected, high, 0.912, but not fully coinciding. This seems to justify the new measure.

**Figure 13 entropy-24-01048-f013:**
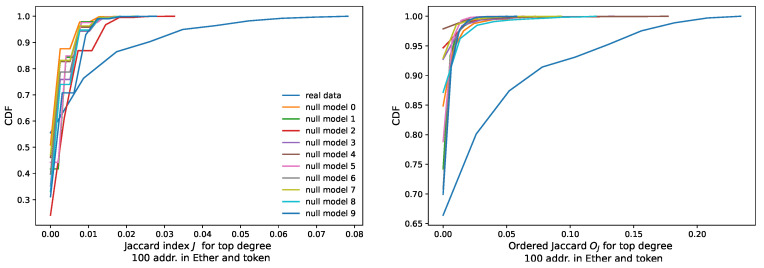
To verify the validity of our proposed similarity metric, we test it against a null model where addresses are randomly drawn with the same frequency that we observe in the real data of 161,000 addresses. We then take every time 2432 buckets of addresses for the tokens plus one additional for the native cryptocurrency Ether. We repeat this random draw 10 times and then compare the resulting null model with the real one in a Cumulative Distribution Function (CDF) plot. We do this for the normal Jaccard Index *J*, and for the Ordered Jaccard Index $O_{j}$. The real data we have are well discernible from purely random curves for both metrics: random data is much more concentrated, while real data follows a much softer slope, with values arriving up to 0.08 for the Jaccard Index *J* and 0.25 for the Ordered Jaccard $O_{j}$, while all simulated attempts displayed are much more skewed towards 0 and stop earlier in the metric. The difference, not present in other tested metrics, strongly suggests that we are measuring with both the Jaccard and the Ordered Jaccard Indexes a fundamental property of our data set and not just a random number. Additionally, the Ordered Jaccard Index seems to distinguish clearer between real data and null models.

**Figure 14 entropy-24-01048-f014:**
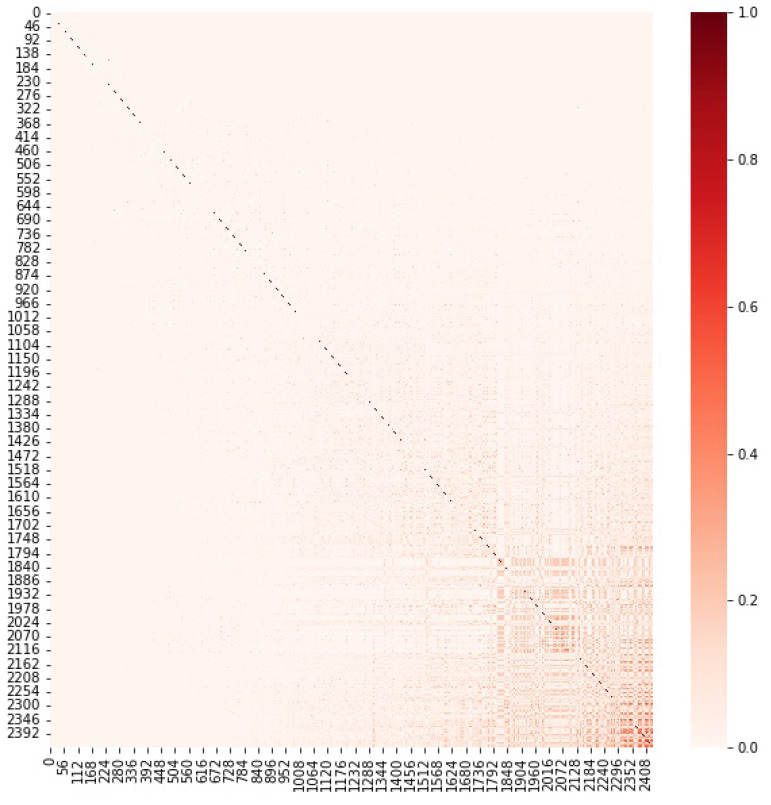
Heat map with the Ordered Jaccard Index $O_{j}$ between all 2432 tokens. Tokens have been ordered according to their average $\langle O_{j}\rangle$ with all the other tokens.

**Figure 15 entropy-24-01048-f015:**
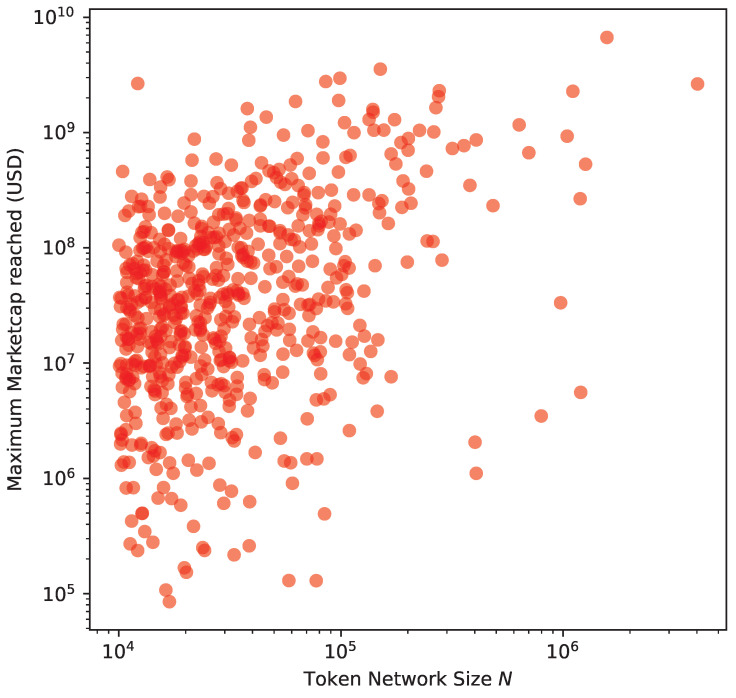
Network size as number of addresses *N* and maximum market capitalisation reached in USD. Since we are dealing with cumulative networks, the maximum market capitalisation ever reached compounds for all the previous network growth. We obtain a Pearson correlation ρ of 0.3318.

**Figure 16 entropy-24-01048-f016:**
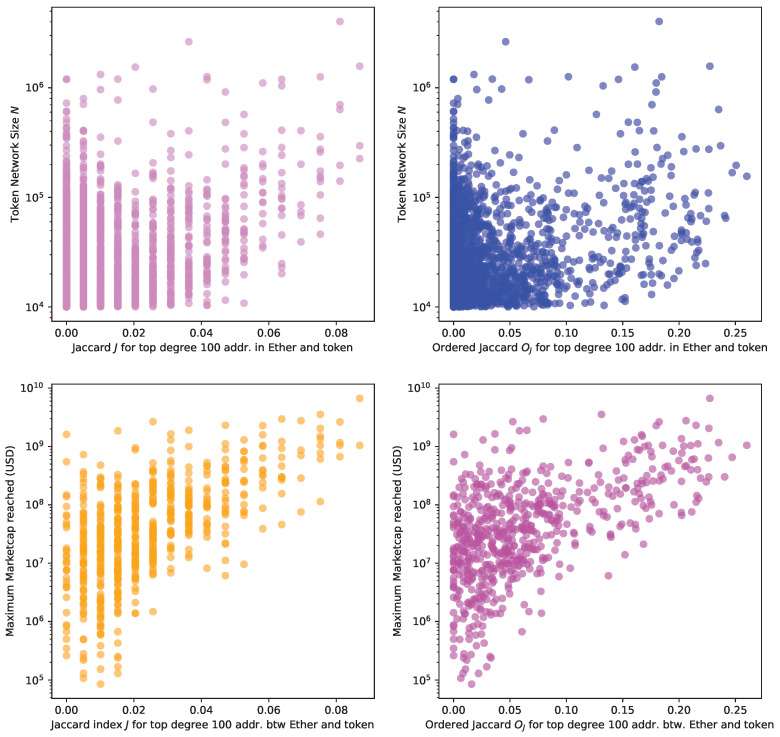
We plot the Jaccard Index *J* and the Ordered Jaccard Index $O_{j}$ between the 100 highest-degree nodes in “greedy” dismantling order between the individual token and Ether against the network size *N* (**top** plots) and the maximum market capitalisation ever recorded in the token history (**bottom** plots).

**Figure 17 entropy-24-01048-f017:**
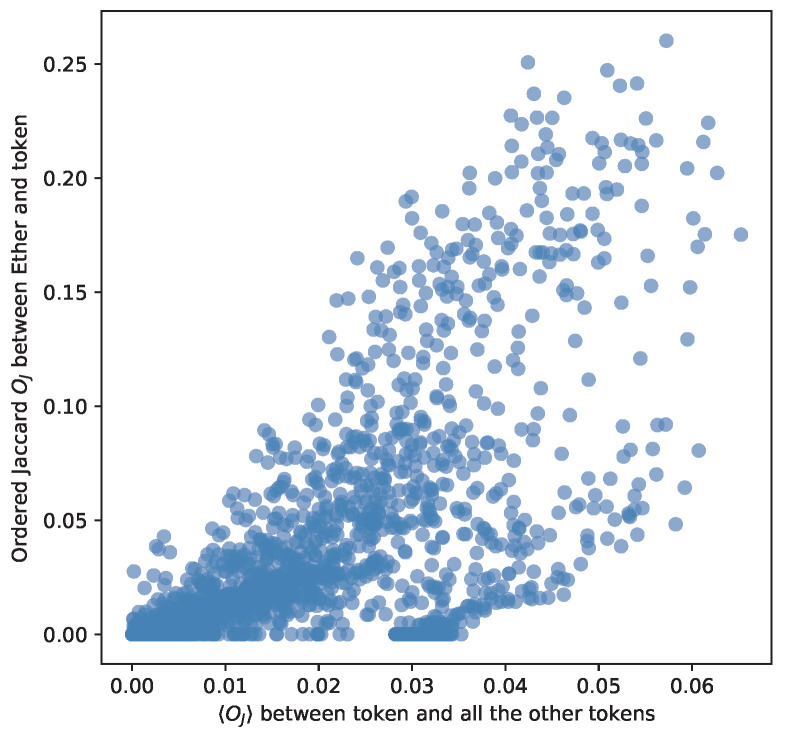
We plot the Ordered Jaccard Index $O_{j}$ between Ether and a specific token with the average $\langle O_{j}\rangle$ for that token with all other Ethereum-based tokens. The measured correlation Pearson correlation ρ reaches 0.674613. It confirms the relevance of this metric as a general measure of similarity between tokens.

**Figure 18 entropy-24-01048-f018:**
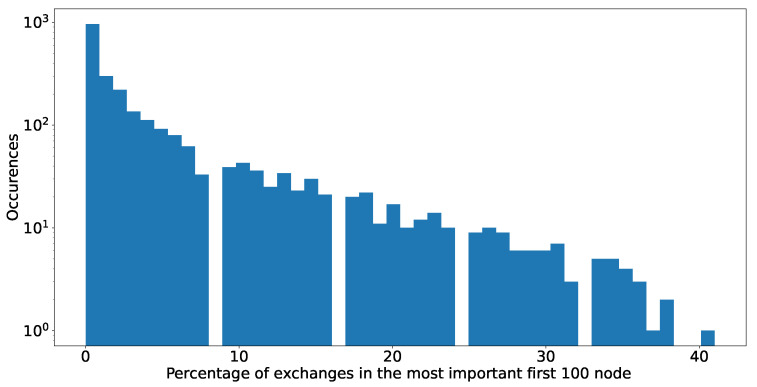
Number of first 100 higher degree nodes in greedy dismantling order that can be labelled as exchange according to information from etherscan.io.

**Figure 19 entropy-24-01048-f019:**
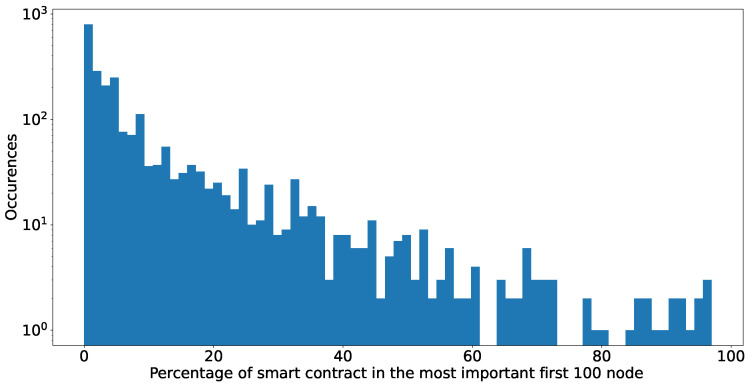
Number of the first 100 nodes of every token network are actually smart contracts.

**Figure 20 entropy-24-01048-f020:**
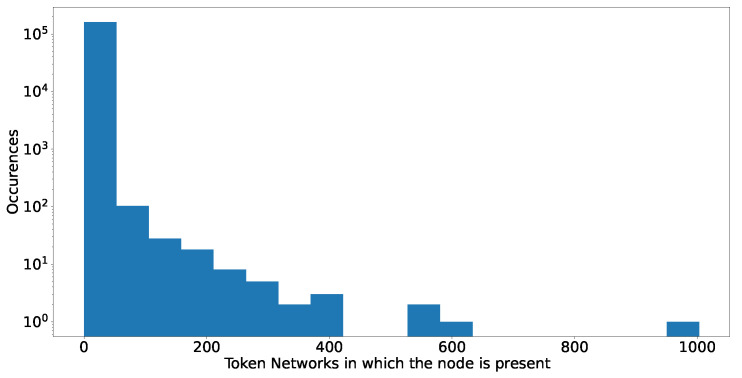
Reappearance frequency of dismantled nodes between different token networks (plus Ether), i.e., in how many different networks the individual addresses are appearing.

**Table 1 entropy-24-01048-t001:** Summary of the data sets used in this analysis, including block range and time span for each token. These cover the time span since the creation of each token, with the exception of DAI. The launch of DAI took place in 2017, but we only use transactions since its transition to a multi-collateral model in 2019.

Token	Tx	Nodes	Edges	Blocks	Time Span
AMP	755,827	83,050	201,456	7,953,823–12,500,000	14 June 2019–25 May 2021
BAT	3,046,615	1,105,958	1,702,429	3,788,601–12,500,000	29 May 2017–25 May 2021
DAI	8,422,158	1,042,638	2,523,076	8,928,674–12,500,000	13 November 2019–25 May 2021
UNI	2,079,132	701,054	1,271,933	10,861,674–12,500,000	14 September 2020–25 May 2021

**Table 2 entropy-24-01048-t002:** Spearman correlation $\rho_{s}$ between in-degree $k_{in}$ and out-degree $k_{out}$ for the four tokens. The relation is stronger for AMP and DAI than for BAT and UNI. However, we observe an irregular pattern for all nodes. We suspected that the correlation for nodes with degrees over 100 could even be stronger. Computing the Spearman correlation for $k_{in}>100$ confirmed this point.

Token	$\rho_{s}\left(k_{\mathrm{in}},k_{\mathrm{out}}\right)$	*p*-Value	$\rho_{s}\left(k_{\mathrm{in}},k_{\mathrm{out}}\right)$ Where $k_{\mathrm{in}}>100$	*p*-Value
AMP	0.5201	0	0.6772	1.2470 $\times \;10^{-7}$
BAT	0.1523	0	0.4119	2.6450 $\times \;10^{-13}$
DAI	0.4842	0	0.4874	5.112 $\times \;10^{-48}$
UNI	0.2710	0	0.5094	1.3512 $\times \;10^{-15}$

**Table 3 entropy-24-01048-t003:** Power law fit for the in- and out-degree distributions of the four tokens. The exponent γ of the power law degree distribution $p_{k}∼k^{-\gamma }$ typically fulfils 2≤γ≤3 for the network to be characterized as scale-free [35,37]. This condition occurs only for the in-degree $k_{in}$ of AMP and out-degree $k_{out}$ of DAI, which suggests that for most of our cases the conditions for a scale-free network are weak [38]; $x_{min}$ is the minimum *x* value where the fit starts. The table also includes the standard error σ for all coefficients to facilitate the assessment of the fit.

Token	*k*	$x_{\min}$	γ	σ	d	Best Fit
AMP	$k_{in}$	3.0	2.9254	0.0169	0.0362	Power Law
AMP	$k_{out}$	13.0	1.6150	0.0409	0.0395	Power Law
BAT	$k_{in}$	44.0	1.7677	0.0330	0.0292	Power Law
BAT	$k_{out}$	58.0	1.6580	0.0326	0.0304	Truncated Power Law
DAI	$k_{in}$	57.0	1.8552	0.0240	0.0115	Power Law
DAI	$k_{out}$	4.0	2.5021	0.0055	0.0121	Lognormal
UNI	$k_{in}$	51.0	1.7812	0.0409	0.0271	Power Law
UNI	$k_{out}$	29.0	1.6591	0.0299	0.0300	Power Law

**Table 4 entropy-24-01048-t004:** The $\alpha^{in}$ and $\alpha^{out}$ for each of the four analysed tokens and their errors. All values of α are higher than 1, indicating a super-linear preferential attachment.

Token	$\alpha^{\mathrm{in}}$	Error	$\alpha^{\mathrm{out}}$	Error
AMP	1.05	0.143	1.02	0.174
BAT	1.15	0.198	1.1	0.226
DAI	1.1	0.099	1.05	0.126
UNI	1.05	0.227	1.02	0.257

## Data Availability

Public Blockchain data are inherently public: all the data used is therefore available in the Ethereum blockchain. As clients, both OpenEthereum and Erigon were used for performance reasons. Data were transformed with ethereum-etl tool for further processing. The data set is publicly available on Google Cloud BigQuery as in [52], where it can be queried directly from a web interface. The technical architecture of the solution connecting Google Cloud and ethereum-etl is well described in [53]. For intensive analysis (such as the one conducted in this paper) a local extraction of the data is warmly recommended. Internal transaction data are not stored on chain, therefore we extracted them from the public API of Etherscan [54].

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
