# Peer review of "The Structural Role of Smart Contracts and Exchanges in the Centralisation of Ethereum-Based Cryptoassets"

_entropy, 2022, doi:10.3390/e24081048_

Round 1

Reviewer 1 Report

The paper written by Francesco Maria De Collibus et al. analyses structural properties of transaction networks in Ethereum-based tokens. In this paper, they analyze the transaction networks of four DeFi-collateral tokens and extend this analysis to all Ethereum-based tokens. Besides, they assess the structural roles of smart contracts and externally owned accounts. The results show that smart contract and exchange-related addresses play a crucial role in holding together the transaction networks. In general, this paper has moderate originality.

There are some details in the article that are not clearly introduced.

1) This paper needs further proofreading because of many language errors, such as singular and plural problems and redundant tying verbs. Readers usually have difficulty with the thought of the paper. I suggest that it might be helpful to hire a professional English editor to polish the language of the paper after revising the content.

2) There are too many keywords.

3) On the whole, the manuscript seems more a technical report than a scientific contribution at this moment. In the introduction, I do not think the authors make it very clear of the research gaps and scientific contributions. Few existing studies of Ethereum-based tokens are mentioned. Two important question shall be answered: why do you analyse structural properties of transaction networks in tokens running on the Ethereum blockchain? Does this work fill up some knowledge gaps which previous articles cannot address?

4) The author should provide the advantages of the proposed Ordered Jaccard Index to other types of metrics.

5) The final part should be more refined to make the findings and contributions of the paper clearer. In addition, please strengthen the explanation of their significance.

6) The overall structure of the article is not good. The authors divide experimental analysis into two parts, but it is only mentioned in the discussion section. Readers cannot distinguish it from the overall structure.

7) Figures must come immediately after the first time it is mentioned in the text, not far away.

Reviewer 2 Report

The manuscript is devoted to a study of a topic of a research area which is hot in the last years: the blockchain technologies. The Ethereum blockchain is analyzed from the point of views of the properties of transaction networks in tokens running on this blockchain Ampleforth (AMP), Basic Attention Token (BAT), Dai (DAI) and Uniswap (UNI). For blockchain nodes of kinds smart contracts (SC),  and externally owned accounts (EOA), authors study their structural role and importance for the network structure. This study is done by quantities which are typical for network studies and by study for existence of scenarios such as preferential attachment. There is a methodological new point in the manuscript: the authors propose a version of the Jaccard index which version is called Ordered Jaccard Index.  The main conclusion of the study is that the two kinds of nodes:  smart contract and exchange-related addresses play an important structural role in transaction networks and are connected to the success or to the failure of the network and are also very important for   the centralization of the decentralized finance (DeFi) ecosystem. The analyzed data are original and the used methodology for analysis is very appropriate.  The presentation is clear and the figures provide appropriate illustration of the text.  Because of this my opinion about the publication of the manuscript is positive.

Author Response

Thank you so much for your kind feedback! We have appreciated it very much! 

Reviewer 3 Report

ENTROPY_The structural role of smart contracts and exchanges in the centralisation of Ethereum-based cryptoassets

Special Issue: Selected Papers from the Tenth International Conference on Complex Networks & Their Application

…………………………

In this paper the authors, analyze 4 Ethereum-based tokens through the Ether network with the purpose of studying the most important in the both the ETH and the token transaction networks.

The study is very interesting, and the results are of high importance. The paper is well-structured, well-written and justified. It is almost ready for publication.

1). Page 4. The authors write: “market capitalisation data from a popular site with price quotations, Coinmarketcap ([21]). We select a set of a hundred nodes with the highest degree. This threshold is certainly arbitrary, since we are considering networks of different size, however this value proved reasonable in our deep-dive”. Even if arbitrary, as they mention, it needs to be explained.

2). Page 24. “Our analysis of the token market capitalization indicate that investor interest makes networks grow in size, which eventually increases the market capitalization of the token as well”. Given that the paper utilizes PA methods, I find this as a tautology. Please explain.

3). Page 24. “It is obvious is that this “auto-mated engine” of smart contracts… “. Not clear.

4). The authors chose a model based on preferential attachment (PA). I would suggest the authors to justify their choice and explain why PA fits the problem.
